# Inheritance-Specific Dysregulation of Th1- and Th17-Associated Cytokines in Alopecia Areata

**DOI:** 10.3390/biom13091285

**Published:** 2023-08-23

**Authors:** Monica M. Van Acker, Rebekah R. Schwartz, Kelly Andrews, Kristina Seiffert-Sinha, Animesh A. Sinha

**Affiliations:** Department of Dermatology, Jacobs School of Medicine and Biomedical Sciences, University at Buffalo, Buffalo, NY 14203, USA

**Keywords:** Th1, Th17, IL-17A, IL-23, IFN-gamma, TNF-alpha

## Abstract

Autoimmune diseases tend to cluster in families, suggesting genetic predisposition to autoimmunity associated with familial background. We have previously reported similarities in gene expression patterns and PTPN22 polymorphisms between alopecia areata (AA) patients and their healthy relatives, but not unrelated healthy controls. However, the spectrum of disease promoting (or preventing) pathways that may be activated in blood relatives of AA patients remains to be defined. Here, we investigated the extent to which cytokines associated with the Th1 and Th17 pathway are differentially expressed in the blood of patients with AA and its clinical subtypes in comparison to both healthy relatives as well as unrelated healthy controls. A comprehensive set of Th1- and Th17-related cytokines were evaluated by ELISA. We found a significant elevation of the Th17 inducer IL-23, the Th17 product IL-17A, the Th1 hallmark cytokine IFNγ, and TNFα, a Th1 cytokine with relevance to the Th17 pathway in AA patients, regardless of disease subtype, compared to healthy individuals. On further examination, we found that healthy family members grouped together with patients in terms of elevated Th1- and Th17-pathway cytokines in an inheritance-specific manner, distinct from unrelated controls. The elevation of Th17-associated cytokines in healthy controls related to AA patients indicates that Th1 and Th17 dysregulation in AA may be genetically based. Of note, one unrelated control displayed elevated levels of IL-17A and IL-23 similar to those detected in patients. One year after initial blood draw, areas of beard hair loss consistent with the diagnosis of AA were reported by this individual, indicating that the elevation in Th17-related cytokines may have predictive value.

## 1. Introduction

Alopecia areata (AA) is a nonscarring autoimmune hair loss condition with a prevalence of 1.7% in the general population [1]. For the majority, AA will not progress beyond a limited number of well-circumscribed patchy areas that are self-limited and reversible. But for an estimated 7–10% of the AA population, the development of chronic disease, with total hair loss of the scalp (alopecia totalis) or the entire body (alopecia universalis), will ensue [2]. It is well accepted that the interplay of genetics and environmental triggers is relevant to disease onset and development [3,4,5]. Still, the etiopathogenesis behind AA remains elusive.

Histopathologically, lesional biopsies of AA patients are characterized by the presence of a perifollicular and intrafollicular T cell infiltrate (predominantly a CD4+ T helper (Th) 1, but also CD8+ cells) along with Langerhans cells, macrophages, and NK cells [3,6]. Along with the predominant lesional CD4+ cell infiltrate, AA patient sera and lesional biopsies display Th1 pathway cytokines such as IFNγ, IL-2, IL-12, and TNFα [7]. On the other hand, AA is frequently affiliated with atopic diseases seemingly derived from a Th2 pathway response [8]. It has been hypothesized that Th1 cytokines may be crucial in disease progression, while Th2 cells play a role in preventing or delaying further disease progression [9]. Although there remain considerable gaps in the knowledge regarding the precise roles Th1 and Th2 cytokines play in AA, the systemic cytokine milieu is likely to be critical in the generation and regulation of the autoimmune response that is directed to the hair follicle [10]. Given the major pathophysiologic contribution of T cells [11], the disease mechanisms in AA can be expected to involve both a direct and supporting role for lymphocyte-derived cytokines.

In the past, AA was considered to be a Th1-driven disease, but, in the past two decades, the Th17 T cell pathway has gained attention for its impact on the immune system. Th17 cells are identified by their secretion of the proinflammatory cytokine IL-17A, which has the ability to induce tissue injury correlated with autoimmunity [12]. Th17 cells have been shown to be relevant in several autoimmune diseases including inflammatory bowel disease, multiple sclerosis, psoriasis, eczema, rheumatoid arthritis, Sjögren’s syndrome, and vitiligo [13,14,15]. Specifically in AA, several studies on the role of Th17 cells in AA have shown increased serum levels of IL-17 in AA patients when compared to controls [15,16,17,18,19,20,21,22,23,24,25,26,27,28] (Table 1), and skewing towards the Th17 pathway [19,29]. These findings are supported by studies indicating IL-17 activation in a murine model of alopecia [30]. Additionally, genome-wide association studies have identified numerous susceptibility loci, including a region on chromosome 4q27 containing both IL-2 and IL-21 encoding genes, representing a major inflammatory product and a key promoter of the differentiation of Th17 cells, respectively [31,32,33]. Further, other genome-wide association studies have implicated other genotypes of IL-17A and IL-23 that are associated with AA [34,35]. However, a gene expression analysis of lesional vs. non-lesional scalp samples from AA patients found no skewing towards a Th17/Th22 phenotype in lesional skin [36]. Table 1 lists the cytokines that have been studied and found to be dysregulated in the context of AA, with many—but not all—studies showing increases of multiple Th1, Th2, and Th17 pathway-associated cytokines.

Of note, none of the previous studies stratified their control population based on their status of blood relation to AA patients. Given the high twin concordance rate in AA and the fact that autoimmune diseases in general, and alopecia areata in particular, tend to cluster in families, a genetic predisposition to autoimmunity associated with familial background has been suggested. Our group has previously shown that in terms of gene expression, AA patients and individuals related to AA patients are distinct from healthy unrelated individuals [10]. Furthermore, we found that autoimmune-disease-associated polymorphisms of the PTPN22 gene are represented in AA patients as well as healthy relatives, but not in unrelated controls [37]. Similarly, in other autoimmune diseases, such as Pemphigus vulgaris, we have found that healthy controls that are family members of PV patients or carry certain PV-associated HLA susceptibility loci cluster with patients and distinct from non-family member controls [38]. Thus, genetic mechanisms common to patients with autoimmune disease as well as their healthy relatives could be relevant across diseases.

To date, it is unknown whether the potential T helper cell pathway associations in AA extend to healthy family members. To address this gap in the knowledge, we evaluated the levels of Th17-promoting cytokines and cytokines produced by Th17 cells (Appendix A) in the blood of AA patients by ELISA and compared them to unaffected relatives as well as healthy unrelated subjects. Because the fate of T cell differentiation is determined by the cytokine milieu present at activation and because of the reciprocal nature of different T cell subsets, we also investigated key Th1 and Treg pathway cytokines.

**Table 1 biomolecules-13-01285-t001:** Th1-, Th2-, and Th17-associated cytokines in AA reported in the literature.

	Cytokine	Number of Studies Showing Increase	Number of Studies with No Significant Change	Number of Studies Showing Decrease	Polymorphism	Clinical Correlation/ Treatment Response	Present Study
Th17	IL-17	11 [15,16,18,19,20,21,22,24,25,26,27]	3 [39,40,41]		IL17 GG genotype (rs763780; A7488G) [34]	IL-17 consistently elevated; effective treatments have effects on IL-17 [42] Targeting IL-17 may indirectly reduce IL-23 expression [43] Activation coincides with flares; IL-17 as driver of inflammasome cascade [44] 4 case reports of pts treated with inhibitor developing AA [45]	
	IL-17A	6 [17,19,20,22,23,26]				No response to secukinumab [46]	Increased
	IL-21	2 [15,19]			Genomic regions for activation associated with AA [33]	Janus kinase inhibitors [47]	No significant change
	IL-22	2 [15,24]	2 [24,40]				No significant change
	IL-23	2 [19,48]	2 [22,24]		AA genotype of *IL23R* (rs10889677) [35]	No response to ustekinumab [49] Efficacy of ustekinumab may be mediated by IL-23 antagonism [43]	Increased
	IL-17F	1 [19]					
Th1	IFN-γ	13 [9,22,24,25,26,27,40,50,51,52,53,54,55]	2 [19,24]			Blockade prevents disease, ruxolitinib—hair regrowth [56] JAK inhibitors [43]	Increased
	IL-2	8 [9,17,18,22,27,39,52,53]	1 [40]	1 [26]	Genomic regions for activation associated with AA [33]	Janus kinase inhibitors [47] Blockade prevents disease [56]	
	IL-12	2 [40,53]			*CC genotype IL12B* (rs3212227) [35]	No response to ustekinumab [49] Downregulation with corticosteroid treatment [32]	
	TNF-α	7 [15,18,19,24,48,52,55]	1 [53]			Case report; worsening AA with infliximab [57] No improvement with TNF-α antagonists [58]	Increased
Th2	IL-4	4 [9,25,39,59]	3 [18,40,53]	1 [22]	Gene intron 3 VNTR polymorphism associated with increased risk of AA [60]	Role for IL-4Rα antagonist (implicated in atopic dermatitis, which has similar genetic framework to AA) [43]	
	IL-5		1 [18]	1 [40,61]			
	IL-13	3 [25,27,51]	2 [40,53]	1 [24]	rs20541 (IL-13) susceptibility loci in AA [62]	Potential role for dupilumab; increased mRNA in lesional vs. non-lesional scalp [43]	
other	IL-1	1 [24]					No significant change
	IL-6	6 [15,19,26,48,50,53]	1 [63]				No significant change
	IL-7	1 [64]				Janus kinase inhibitors [47]	
	IL-9	1 [25]	1 [53]				
	IL-10	3 [19,22,53]	4 [17,27,40,63]	1 [65]			No significant change
	IL-15	2 [26,39]				Janus kinase inhibitors [47] Blockade prevents disease [56]	
	IL-16				Exon rs11073001 promoter region rs17875491 [66]		
	IL-18	1 [67]	2 [40,53]				
	IL-25	1 [19]					
	IL-31	1 [19]					
	IL-32					Downregulation with corticosteroid treatment [32]	
	IL-33	1 [19]					
	TGF-β	1 [24]		2 [18,27]			No significant change
	CXCL1				rs3117604, −429C/T [68]		
	CXCL2				rs3806792, −264T/C [68]		
	CCL-18					Downregulation with corticosteroid treatment [32]	

## 2. Methods

### 2.1. Patient Recruitment

The study was approved by the Institutional Review Boards of Weill Medical College of Cornell University (IRB No. 1098-429) and Michigan State University (IRB No. 05-1026). Informed consent was obtained from all patients before venous blood collection. Sixty-four patients diagnosed with AA were enrolled for this study at meetings of the National Alopecia Areata Foundation (NAAF), and at the dermatology outpatient clinics at New York Hospital and Michigan State University. At the time of enrollment, disease severity, sex, and age were recorded for each patient (Table 2). The study participants were mainly of Caucasian descent (*n* = 72), followed by African American (*n* = 10), Hispanic (*n* = 8), Asian (*n* = 4), and Southeast Asian (Indian) (*n* = 2). Patients with alopecia were divided into clinical subtypes based on the NAAF classification: AA transitory (AAT; loss of hair in patches on the scalp <1 year), AA persistent (AAP; loss of hair in patches on the scalp >1 year), Alopecia totalis (AT; complete loss of scalp hair with or without body hair loss), and Alopecia universalis (AU; loss of hair over the entire body). Of the alopecia patients surveyed, 16 were classified as AT (5 males and 11 females), 16 as AU (5 males and 11 females), 15 as AAT (4 males and 11 females), and 17 as AAP (5 males and 12 females). Additionally, 16 healthy unaffected relatives (UR) of AA patients and 16 healthy blood donors (UNR) with no family history of autoimmune disease were enrolled in this study. All unaffected relatives were 1st degree relatives (i.e., parents, full siblings, or children). None of the patients or controls were receiving systemic immunomodulating treatment at the time of the blood draw. One of the control subjects in the UNR group later developed AA. This individual remained in the UNR group and was excluded only for a re-analysis of the data, as described in the Section 3. 

### 2.2. Detection of Serum Cytokines

A comprehensive set of Th17-related cytokines was evaluated by ELISA: (i) cytokines that promote Th17 differentiation in naive T cells (IL-1β, IL-6, IL-21, IL-23, and TGFβ1), and (ii) cytokines produced by Th17 cells (IL-17A, IL-21, IL-22) [69,70,71,72] (Appendix A). Additionally, we evaluated the Th1 hallmark cytokines IFNγ and TNFα and the T regulatory (Treg) hallmark cytokine IL-10. All cytokines except IL-22 were evaluated using ELISA Ready-Set-Go! kits (eBioscience, San Diego, CA, USA) according to the manufacturer’s instructions. IL-22 was evaluated in a separate ELISA kit purchased from Antigenix America, Huntington Station, NY. All serum samples remained undiluted during the evaluation of cytokine levels except for the evaluation of TGFβ1 levels in which serum was diluted 1:5. In short, the ELISA plates were initially incubated with cytokine-specific capture antibody, followed by the addition of patient serum, and finally the addition of cytokine-specific detection antibody. The binding of the detection antibody was visualized by adding avidin-conjugated horseradish peroxidase and an enzyme-specific substrate. Specific cytokine quantities were measured using a Multiskan FC Microplate Spectrophotometer (Thermo Scientific, Waltham, MA, USA) and compared to manufacturer-supplied cytokine standards (presented in pg/mL).

### 2.3. Statistical Analysis

The categorical evaluation and comparison of groups was based upon (i) subtype-specific, (ii) disease-specific, and (iii) inheritance-specific differences. To analyze whether differing levels of Th17-associated cytokines affect the severity of disease (i.e., subtype-specific differences), we compared clinically mild forms of AA (AAT and AAP) with clinically severe forms (AT and AU). To analyze whether the presence or absence of disease was reflected in AA-associated cytokine levels (i.e., disease-specific differences), we compared patients with all subtypes of AA (AAT, AAP, AT, and AU) to unaffected relative controls (UR) for each cytokine. Finally, to evaluate a potential genetic basis of AA-associated cytokine dysregulation (i.e., inheritance-specific differences), all subtypes of AA and unaffected relatives (AAT, AAP, AT, AU, and UR) were compared to unaffected unrelated controls (UNR) for each cytokine. The statistical analysis of serum cytokine levels among individual groups and subtype-specific, disease-specific, and inheritance-specific differences were calculated using the Kruskal–Wallis nonparametric comparison method for analyzing population medians among unpaired groups [73]. Statistical significance was recognized at *p* < 0.05. Additionally, a heat map was used to visualize differential levels of log-transformed cytokine serum levels implemented in the Partek Genomic Suite v6.6 (Partek, St. Louis, MO, USA). The findings are clustered based upon well-defined similarity rules.

## 3. Results

### 3.1. Significant Inheritance-Specific Upregulation of Th17-Produced Cytokine IL-17A Is Revealed in AA Patients and Their Healthy Relatives vs. Unaffected Non-Relative Controls

We first evaluated the cytokines produced by Th17 cells, including IL-17A, IL-21, and IL-22. Among the cytokines produced by Th17 cells, IL-17A is considered the hallmark cytokine associated with Th17 function [74]. The results show that 80 out of a total of 96 study subjects had detectable levels of IL-17A protein with concentrations ranging from 0.06 to 32 pg/mL (4.34 ± 6.30 pg/mL, median 2 pg/mL). IL-17A: AAT = 5.21 pg/mL ± 7.54, AAP 4.06 pg/mL ± 3.95, AU = 5.56 pg/mL ± 9.24, AT = 7.55 pg/mL ± 7.81, UR = 3.42 pg/mL ± 1.48, UNR = 0.32 pg/mL ± 1.21. There were no significant differences in the cytokine levels between AA clinical subtypes based on NAAF classifications (AAT vs. AAP vs. AT vs. AU; *p* = 0.141). When comparing AA patients based on disease severity, there were no significant differences displayed between mild (AAT+AAP) vs. severe (AT+AU) forms of AA (*p* = 0.237). However, when all (unstratified) patients with AA were compared to unstratified controls (healthy unaffected relatives of AA patients and healthy non-relatives), a significant upregulation of IL-17A protein levels in the patient group was observed (*p* = 0.001). A further examination of the data indicated that the overall IL-17A levels in the unaffected relatives of AA patients (UR) were similar to the levels seen in AA patients, but distinct from non-relatives (UNR). Thus, when comparing unstratified AA patients (AAT+AAP+AT+AU) to unaffected relatives (UR), there was no significant difference in IL-17A levels (*p* = 0.665). Furthermore, the comparisons of unaffected relatives of AA patients vs. non-relatives, as well as all (unstratified) AA patients and their unaffected relatives (AAT+AAP+AT+AU+UR) grouped together vs. unaffected non-relative controls, revealed a significant upregulation of IL-17A protein levels in the patient-related groups (*p* < 0.001 in both comparisons) (Figure 1A). Our data indicate that there is a significant dysregulation of IL-17A protein levels in an inheritance-specific manner whereby patients, regardless of disease severity, cluster together with their healthy relatives, but not with healthy non-related controls. No significant differences were seen for any of the other cytokines produced by Th17 cells, IL-21, and IL-22, in any of the comparisons.

### 3.2. Significant Inheritance-Specific Upregulation of the Th17 Promoting Cytokine IL-23 Is Revealed in AA Patients and Their Healthy Relatives vs. Unaffected Non-Relative Controls

Next, we evaluated cytokines that promote Th17 differentiation, including IL-1β, IL-6, IL-21, IL-23, and TGFβ1. Among these cytokines, IL-23 plays a pivotal role in the formation and amplification of Th17 cells [75,76]. Here, 61 out of 96 total study subjects had detectable levels of IL-23 protein with concentrations ranging from 0.04 to 66 pg/mL (8.31 ± 13.44 pg/mL, median 2.79 pg/mL). AAT = 13.78 pg/mL ± 20.70, AAP = 6.66 pg/mL ± 8.2, AU = 7.74 pg/mL ± 14.7, AT = 9.38 pg/mL ± 9.32, UR = 11.01 pg/mL ± 16.43, UNR = 1.76 pg/mL ± 3.32. No significant differences in IL-23 protein levels were revealed between any of the AA clinical subtypes (*p* = 0.540), between mild and severe forms of AA (*p* = 0.868), or between all (unstratified) AA patients compared to their unaffected relatives (*p* = 0.669). However, when we compared all (unstratified) patients with AA grouped together with unaffected relatives vs. unaffected non-relative controls (UNR), we saw a trend towards the significant upregulation of IL-23 protein levels in the patient-related group (*p* = 0.069) (Figure 1B). However, in reanalysis of the data following the removal of one subject initially classified as a control, but who later developed AA, the IL-23 comparison reached statistical significance (*p* = 0.038; see below for a more detailed explanation of these data). Thus, we find that for the Th17-promoting cytokine IL-23, patients, regardless of disease severity, cluster in an inheritance-specific manner together with their healthy relatives, but not with healthy unrelated controls. No significant differences were seen for any of the other cytokines involved in the promotion of Th17 differentiation, IL-1β, IL-6, IL-21, and TGFβ1, in any of the comparisons.

### 3.3. No Differences Are Observed for T Reg Cytokines

While Th17 and T reg cells are known to share a common signaling pathway mediated by TGFβ, other proinflammatory signals present during cell activation regulate the fate of these cells reciprocally and the cell types play opposing roles during inflammation [77]. T reg cells inhibit autoimmune responses by producing the anti-inflammatory cytokines IL-10 and TGFβ. However, in this study, we did not see any significant differences between AA patients and their healthy relatives vs. unaffected non-relative controls for IL-10 or TGFβ.

### 3.4. Significant Inheritance-Specific Upregulation of the Th1-Produced Cytokines IFNγ and TNFα Is Revealed in AA Patients and Their Healthy Relatives vs. Unaffected Non-Relative Controls

We additionally evaluated IFNγ, the hallmark Th1 cytokine previously implicated in hair follicle development and AA pathogenesis [78]. We found that 85 of 96 total study subjects had detectable levels of IFNγ protein with concentrations ranging from 0.2 to 425.6 pg/mL (34.65 pg/mL ± 72.26 pg/mL, median 8.8 pg/mL). AAT = 40.46 pg/mL ± 65.55, AAP 12.11 pg/mL ± 12.63, AU = 20.02 pg/mL ± 28.26, AT 80.13 pg/mL ± 115.31, UR = 34.58 pg/mL ± 46.22, UNR 2.11 pg/mL ± 3.80. There were no significant differences in IFNγ levels between clinical subtypes (*p* = 0.054), between mild and severe forms of AA (*p* = 0.171), or between all (unstratified) AA patients compared to their unaffected relatives. However, when we compared all (unstratified) patients with AA to all (unstratified) controls, or all (unstratified) patients with AA together grouped with their unaffected relatives vs. unaffected non-relative controls, we observed a significant upregulation of IFNγ protein levels in the patient-related groups (*p* = 0.004 and *p* < 0.001, respectively (Figure 2A). Thus, similar to IL-17A and IL-23, IFNγ protein levels exhibit a significant dysregulation in an inheritance-specific manner in which patients, regardless of disease severity, cluster together with their healthy relatives, but not with healthy unrelated controls.

Another significant cytokine produced by Th1 cells is TNFα [79] (Appendix A). However, TNFα has also been shown to be of relevance within the Th17 pathway [79] and is associated with the maturation of antigen-presenting dendritic cells that are responsible for the development of a positive feedback loop known to induce the persistent inflammation associated with autoimmune reactions [80,81,82]. Here, 95 out of the 96 subjects in our study had detectable levels of TNFα, ranging from 3.56 to 466 pg/mL (mean 120.84 ± 126.34 pg/mL, median 61.95 pg/mL). TNFα: AAT = 124.08 pg/mL ± 129.98, AAP = 99.29 pg/mL ± 124.84, AU = 143.53 pg/mL ± 129.14, AT = 164.43 pg/mL ± 153.52, UR = 126.75 pg/mL ± 118.82, UNR = 70.45 pg/mL ± 90.48. Similar to our findings for IFNγ and IL-17A, there were no significant differences in the cytokine levels between AA clinical subtypes (*p* = 0.464), between mild and severe forms of AA (*p* = 0.171), or between all AA patients compared to their unaffected relatives (UR) (*p* = 0.904). However, when we compared all (unstratified) patients with AA in a group with unaffected relatives vs. unaffected non-relative controls, we saw a significant upregulation of TNFα protein levels in the patient-related group (*p* = 0.043). Thus, like the previous cytokines, TNFα exhibits an inheritance-specific dysregulation in which patients, regardless of disease severity, cluster together with their healthy relatives, but not with healthy unrelated controls (Figure 2B).

### 3.5. Healthy Control with Elevated IL-17A and IL-23 Cytokine Levels Converted to Clinical AA 12 Months after Initial Blood Draw

In the review of our data, one unaffected non-relative control (control #166) that was included in the IL-17A and IL-23 measurements only displayed noticeably increased levels of IL-17A (4.84 pg/mL) and IL-23 (12.86 pg/mL) in comparison to the other non-relative controls. In fact, the cytokine levels for this control fell in the range of IL-17A and IL-23 levels observed in AA patients and relatives. Interestingly, approximately 12 months after the initial blood draw, this control subject reported the development of a well-circumscribed area of hair loss, measuring approximately 1 × 1 cm in diameter, in his beard. Shortly after, a second area of well-circumscribed hair loss developed in the beard consistent with the diagnosis of AA (diagnosis confirmed by clinical evaluation by a dermatologist). At the time this manuscript was written, the hair had not regrown in the affected areas. In light of this new information, we reanalyzed the comparison of our participants versus unrelated controls excluding this particular study participant from the control group. Of note, this individual was not entirely removed from the study, but rather excluded for just this second analysis. With the exclusion of this control for reanalysis, the comparison of IL-17 levels between AA patients and their unaffected relatives vs. non-relative controls remained statistically significant. Importantly, the levels of IL-23, which had originally only been trending towards significance (*p* = 0.0697), were now found to be significantly higher (*p* = 0.038) in AA patients and their unaffected relatives vs. non-relative controls.

### 3.6. Unbiased Hierarchical Clustering of AA-Associated Cytokines Demonstrates That Unaffected Non-Relative Controls Cluster Distinctly from AA Patients and Their Relatives

The differential levels of all cytokines evaluated in this study were visualized in a heat map using log-transformed cytokine serum levels implemented in the Partek Genomic Suite v6.6 (Partek, St. Louis, MO, USA). Unbiased hierarchical clustering of the cytokine values in patient and control groups revealed that 14 out of 16 unaffected non-relative controls clustered together (Figure 3, boxed in black), but distinctly from AA patients and their healthy relatives. We did not observe any clustering of cytokine profiles in AA patients based on clinical subtype. Sixty-four AA patients, regardless of clinical subtype (AAT, AAP, AT, and AU), and their healthy relatives (UR) showed overlapping cytokine profiles with no distinct separation between groups. Moreover, the cytokine profiles in AA–related controls did not separate from those of AA patients. Of note, the cytokine profile of control #166 (who, as described above, developed AA 12 months post-sampling) did not cluster with other unrelated controls, but rather with unstratified patients. This unbiased clustering approach underscores that, in terms of Th17 pathway-related serum cytokines, healthy relatives of AA patients exhibit a pattern similar to AA patients, but distinctly different from non-related controls.

## 4. Discussion

There is ample evidence suggesting that tissue damage in AA is due to a T cell infiltrate mainly consisting of Th1 cytokine-expressing cells [3,78,83]. Not surprisingly, a majority of studies exploring the serum expression of Th1 cytokines have demonstrated an increase in the Th1 hallmark cytokine IFNγ [9,22,24,25,26,27,40,50,51,52,53,54,55,84], and IFNγ-induced collapse of hair follicle immune privilege has been described as a key element in the pathogenesis of alopecia areata [85]. Additionally, genome-wide transcriptional studies have implicated the presence of increased Th1 type cells in the infiltrate of scalp biopsies in patients with AA [29]. Our finding that IFNγ is significantly increased in the serum of AA patients (regardless of clinical subtype) vs. non-related healthy controls fits well with these prior studies. However, a number of autoimmune diseases previously regarded as Th1-mediated are now being revealed as being driven, at least partially, by the Th17 pathway [12]. A developmental relationship between Th1 and Th17 cells, with Th1-induced IFNγ actually promoting the Th17 pathway, has been suggested [86,87]. There has been some indication of IL-17 activation in a CD8+ driven mouse model of AA [30], and increased serum levels of IL-17 have been found in a number of studies in AA patients when compared to controls [15,16,17,18,19,20,21,22,23,24,25,26,27,31,32,33]. Although the Th17 pathway has continued to be implicated in AA, a genetic underpinning for this has not been established and the potential familial dysregulation of the Th17 pathway in keeping with the autoimmune diathesis seen in families of AA patients has not been assessed to date.

Table 1 summarizes the studies available to us that found changes in Th1- and Th17-associated cytokines as well as cytokines not strictly associated with these pathways in AA patients compared to their healthy counterparts. While most previous studies find an increase in numerous cytokines in AA patients compared to healthy controls, there is also a number of studies finding no significant changes or even decreased serum cytokine levels.

Our findings continue to lend support for the implication of both Th17 and Th1 pathway cytokines in AA, but expand our view to consider the role of these pathways in relatives of AA patients, i.e., subjects that share genetic information with patients. Our study reveals that healthy relatives of AA patients displayed Th1 pathway-related cytokine (IFNγ and TNFα) as well as Th17 pathway-related cytokine (IL-17 and IL-23) levels in a range similar to that of patients that were significantly increased when compared to unrelated controls. Additionally, the unbiased hierarchical clustering of all cytokines evaluated in this study clearly distinguished cytokine profiles of unrelated controls from both AA patients and healthy relatives. These data suggest that there is an inheritance-specific cytokine expression pattern associated with the dysregulation of the Th1 and Th17 pathways in AA. Furthermore, our data reinforce the influential role of genetics in autoimmunity, through the promotion, differentiation and preservation of Th1 and Th17 cells [33]. Our findings also support the novel concept that individuals genetically at risk of developing AA (AA patients and first-degree family members) harbor at least some overlapping immune dysregulation relevant to disease processes.

Several lines of investigation from our lab support this new model of a familial dysregulation pattern in AA, both at the genetic as well as the transcriptional level. Firstly, our group previously undertook a global transcriptional analysis of peripheral blood by microarray to study the gene expression profiles from AA patients as well as their unaffected relatives and unaffected non-relatives. Unbiased hierarchical clustering revealed that AA patients and first-degree relatives of AA patients cluster distinctly from healthy controls with no familial connection to AA, thus producing an “inheritance signature” [10]. Additionally, we previously reported that the protein tyrosine phosphatase N22 (PTPN22) 1858T genotype, a polymorphism that has been linked to T cell suppression, is more frequently seen in AA patients and their primary relatives than unrelated controls [37]. 

While the concept of an inheritance-specific immune dysregulation in AA is new, it is not completely surprising in the context of autoimmunity. A genetic predisposition in AA, with 6–7% of first-degree relatives of individuals affected by AA estimated to develop AA themselves throughout their lifetime, is well accepted [88]. Furthermore, there is a higher prevalence of other autoimmune disease, such as vitiligo, thyroid disease, and collagen vascular diseases, many of which have a humoral component, in AA patients and unaffected relatives [89]. It is intriguing to speculate that there could be a “dilutional” effect in the inheritance-specific Th17 regulation we see in this study based on the degree of genetic similarity, i.e., stronger similarities in 1st-degree relatives than in 2nd- and 3rd-degree relatives. In this study, of the 16 healthy unaffected relatives of AA patients, 13 were 1st-degree (mother, father, or son of an AA patient) and only three were 2nd-degree (sister) relatives. We did not see any statistically significant differences between these two groups for IL17A (*p* = 0.81), IL23 (*p* = 0.54), IFNγ (*p* = 0.25) or TNF-α (*p* = 0.95). However, this comparison is statistically underpowered, and a possible dilutional effect may only be unmasked in studies including larger numbers of 1st-, 2nd-, and 3rd-degree relatives.

In line with the data presented in this manuscript, the functional annotation and pathway analysis of genes differentially expressed in AA patients and their healthy relatives vs. non-relative controls in our earlier gene expression study [10] revealed an upregulation of chemokine/cytokine signaling pathways. In particular, the TNFα gene, which has been implicated in immune function, apoptosis, and AA pathogenesis, was found to be highly upregulated, again supporting our current data showing an upregulation of its gene product, the TNFα protein, in the blood of AA patients and their family members. Interestingly, in addition to being a product of Th1 cells, TNFα can also be secreted by Th17 cells [90], and in a positive feedback loop, TNFα has also been shown to drive monocyte differentiation towards IL-23-expressing dendritic cells that stimulate resting T cells to secrete IL-17 [80]. IL-23 appears to be necessary for the amplification and stabilization of the Th17 lineage [87], although not for initial Th17 differentiation [72]. 

In addition to revealing a novel familial inheritance pattern in AA, the results of this study suggest that examining Th17-related cytokine levels may be of predictive value for disease development. This point is exemplified by a study subject who was initially classified as a healthy, not AA-related control subject and who displayed increased IL-17A and IL-23 levels in the range of AA patients and their healthy family members. This individual developed two lesions in his beard area consistent with the diagnosis of AA 12 months after initial study participation. The change in presentation from control to patient status supports the role of IL-17A and IL-23 in the development of AA, and suggests that Th17 pathway cytokines may be dysregulated even before the development of clinical lesions. At the time of initial analysis and prior to the manifestation of disease, this individual may not have possessed the full complement of (as yet unknown) disease promoting factors to exhibit hair loss. Alternatively, or additionally, counter-regulatory mechanisms both in the healthy relatives of AA patients and AA non-related individuals otherwise predisposed to eventually develop disease may act to prevent disease initiation until undetermined environmental stressors and genetic factors skew the balance of disease promoting vs. regulating factors. Further studies, including longitudinal analyses, will be needed to explore the predictive value of Th17-related cytokines for early diagnosis. Nevertheless, the upregulation of the Th17 pathway cytokines IL-17A, IL-23, and potentially TNFα in AA patients in our study suggests the possibility that targeting the Th17 pathway may be of therapeutic value in AA [91,92,93,94]. Drug therapy specifically targeting the Th17 pathway to treat AA is already in development [43,95]. Limited studies and case reports have explored the off-label use of ustekinumab, an IL-12/IL-23p40 blocker, as treatment for AA, with only some studies demonstrating efficacy [49,96,97]. Additionally, secukinumab, which targets IL-17, was shown not to be effective at treating AA in a randomized control trial [46], but this study was terminated due to low enrollment and had low statistical power. The equivocal results seen thus far in studies exploring the Th17 pathway as a drug target in AA necessitate the development of future larger placebo-controlled trials to better define the impact these cytokines have on the pathogenesis of disease.

While the ultimate target of the autoimmune response in AA is restricted to the hair follicle, the systemic immune environment in AA patients is likely to be critical in the generation and regulation of the inflammatory cascade ultimately leading to hair loss. The data from our group and others clearly support a cellular immune response in lesional skin [98], but there has also been a plethora of data to indicate that humoral mechanisms contribute to AA. Our results suggest that alterations in peripheral blood Th1 and Th17 pathway cytokine expression levels in peripheral blood are not only relevant to AA pathogenesis, but that they follow a familial inheritance pattern consistent with peripheral blood cytokine gene expression data generated by our group earlier [10]. This suggested pattern of heritability and our observation of the conversion of a symptom free control subject with elevated Th17-related cytokines to an AA patient indicates a potential role for Th17-associated cytokines, in particular IL-17A and IL-23, to identify at-risk individuals and links these cytokines to disease development. Thus, our data have direct implications for (1) disease mechanisms, (2) clinical prediction, and (3) the identification of potential new targets for therapy. Further mechanistic studies are clearly needed to more deeply determine the precise role of specified cytokines in disease development. Perhaps most intriguingly, it will be important to determine why genetically similar individuals (1st-degree family members) do not develop disease despite their proclivity towards AA-typical cytokine and gene expression profiles. Family members who do not develop disease must either lack additional and required disease-promoting factors, and/or need “protection” mechanisms to suppress disease expression. In support of the latter possibility, our group previously identified a gene expression “disease protection” signature in healthy individuals that harbor Human Leukocyte Antigen (HLA) alleles linked to the susceptibility of developing the autoimmune blistering disorder Pemphigus vulgaris [38]. We found specific gene signatures that distinguish PV patients from healthy controls that carry PV-associated HLA alleles (HLA-allele matched controls), and distinct signatures distinguishing HLA-matched healthy controls from controls who do not carry the specific HLA genotype associated with PV [38]. Not surprisingly, as for numerous other autoimmune conditions, an earlier genome-wide association study found an association of AA with the HLA region [33]. It remains to be determined whether similar “protective” mechanisms are at play in relatives of AA patients that have activated certain autoimmune pathways without developing disease.

## Figures and Tables

**Figure 1 biomolecules-13-01285-f001:**
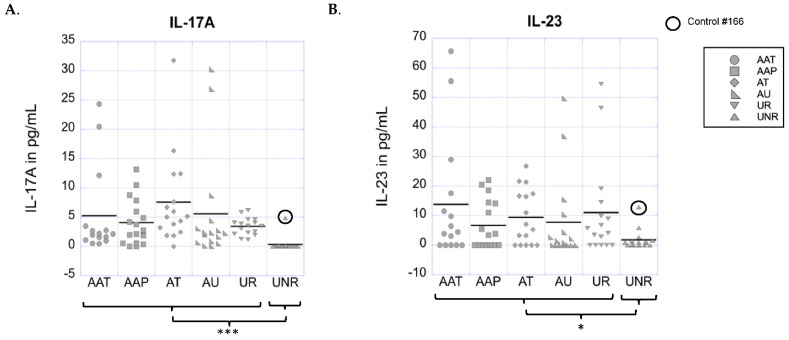
(**A**,**B**). Analysis of cytokines that promote Th17 differentiation and are produced by Th17 cells**.** Serum IL–17A levels (**A**) and serum IL–23 levels (**B**) for individual patients or controls are presented in pg/mL as dot plots. In both plots, mean values for each subgroup are indicated by a horizontal line. The serum value corresponding to control #166 (described in Section 3.5) is indicated with a black circle. * indicates significant differences between groups at a level of *p* < 0.05; *** indicates significant differences between groups at a level of *p* < 0.001.

**Figure 2 biomolecules-13-01285-f002:**
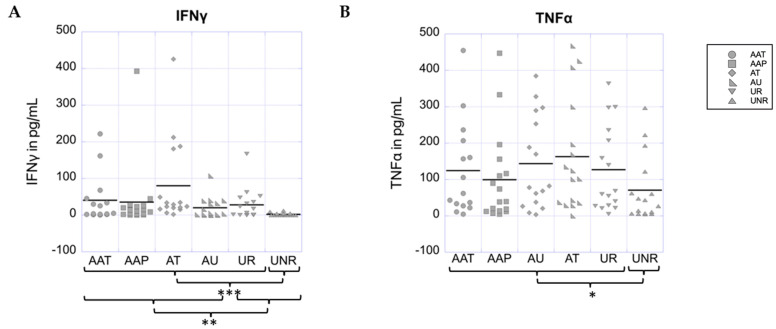
Analysis of the Th1 cytokines IFNγ and TNFα**.** Serum IFNγ (**A**) and serum TNFα (**B**) levels for individual patients or controls are presented in pg/mL as dot plots. In both plots, mean values for each subgroup are indicated by a horizontal line. * indicates significant differences between groups at a level of *p* < 0.05; ** indicates significant differences between groups at a level of *p* < 0.01; *** indicates significant differences between groups at a level of *p* < 0.001.

**Figure 3 biomolecules-13-01285-f003:**
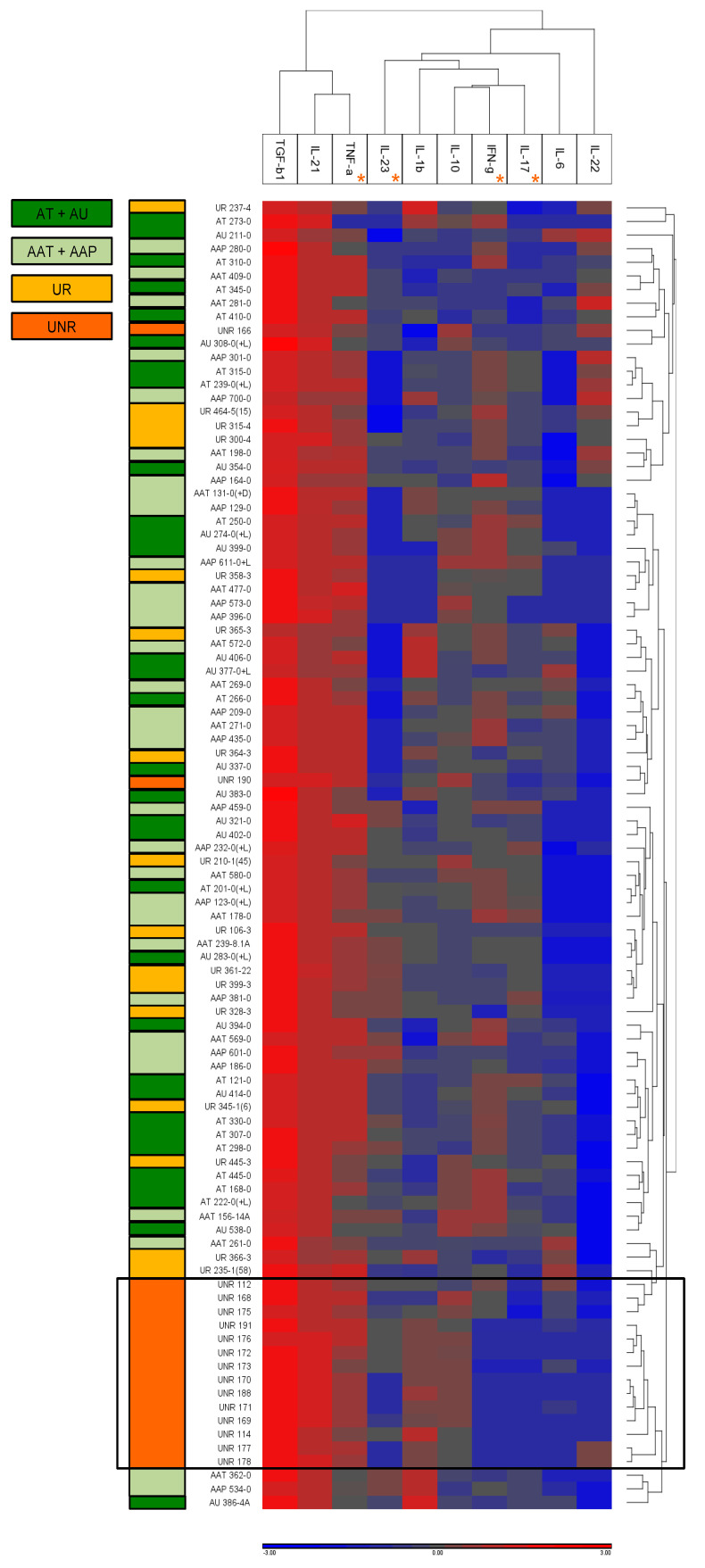
Unbiased hierarchical clustering of Th1, TH17, and T reg cytokines of relevance to AA**.** Log-transformed serum cytokine levels for IFNγ, IL-1β, IL-6, IL-10, IL-17A, IL-21, IL-22, IL-23, TGFβ1, and TNFα for all 96 subjects (64 AA patients, 16 unaffected relative controls, and 16 unaffected non-relative controls) were visualized in a heat map (Partek Genomic Suite v6.6; Partek, St. Louis, MO, USA). Cytokines showing significant differences between patient and control groups by Kruskal–Wallis analyses are marked by a star *. Color legends on the *y*-axis indicate designation for patient and control groups, as follows: light green, mild AA (AAT, AAP); dark green, severe AA (AT, AU); yellow, unaffected relatives (UR); dark orange, unaffected non-related controls (UNR). Within the heat map, red fields represent detection of high, and blue fields detection of low log-transformed levels of a given cytokine. Moreover, 14 out of 16 unaffected non-relative controls cluster together in terms of cytokine profiles (boxed in black), distinct from patient and related control samples. Sixty-four AA patients, regardless of clinical subtype (AAT, AAP, AT, and AU), and their healthy relatives (UR) show overlapping cytokine profiles with no distinct separation between groups.

**Table 2 biomolecules-13-01285-t002:** Study subjects.

Classification of Study Subjects		*n*	Sex		Age (year)	
			Male	Female	Mean Age ± St. Deviation	Range
Alopecia Areata (AA)	All AA	64	19	45	36 ± 17.78	2–71
	Alopecia Totalis (AT)	16	5	11	27 ± 13.47	5–45
	Alopecia Universalis (AU)	16	5	11	44 ± 18.69	11–66
	Alopecia Areata Transitory (AAT)	15	4	11	37 ± 20.81	5–71
	Alopecia Areata Persistent (AAP)	17	5	12	34 ± 14.26	2–58
Controls	Unaffected Relatives (UR)	16	4	12	41 ± 17.84	6–68
	Unaffected Non-Relative (UNR) Controls	16	9	7	31 ± 8.16	27–48

## Data Availability

The data presented in this study are openly available in the Mendeley Data Repository at: Schwartz, Rebekah; Seiffert-Sinha, Kristina (2023), “Biomolecules 2527735 Van Acker et al Cytokines in AA Raw Data”, Mendeley Data, V1, https://doi.org/10.17632/3vvmghjwt6.1 (accessed on 10 July 2023).

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
