# Peer review of "Inheritance-Specific Dysregulation of Th1- and Th17-Associated Cytokines in Alopecia Areata"

_biomolecules, 2023, doi:10.3390/biom13091285_

Round 1
Reviewer 1 Report
In this article, the authors have analysed Th1, Th17 and Tregs assoicated cytokines in a well defined patient group with alopecia, their healthy relatives and controls. I find the hypothesis and aim of study very sound, the introduction is nicely written and the method is also adequate. However, I would have wanted more analyses to support the conclusion, maybe with flow cytometry or similar on immune cells from these individuals, or at least some of the individuals. Also, they could have stimulated cells and investigated immune profiles. It is not surprising that patients with autoimmune disease have an imbalanced immune system, and although I think what was done here was neatly done and presented, I find it a little "scientific light". The discussion is also written in a good way, but I find to be "too large" for this study, which should have been more a brief report. This is my major concern. Other than that, I admire the very nice patient cohort that the group has access to.
Minor concerns:
-Please perform cosmetic changes to Table 1 (i.e write out words, explain abbreviations, be consistent regarding genetic annotation, etc.)
-Do not report the results of the study in the Introduction
-Do not write results in Fig legends. Here, it should only be explained what the Fig. describes.
-Line 131: Lack of "alpha"
-The Figures: Would like to see all points in graphs, not just the mean/median
-Discussion too long for this kind of study.
-Fig 3: Too small text to read.
Author Response
Thank you very much for your time reviewing this paper and for leaving us comments. We have gone through everything you recommended and addressed each concern individually below. All changes within the text are colored in red for your convenience, which will be removed later.
Concern: “However, I would have wanted more analyses to support the conclusion, maybe with flow cytometry or similar on immune cells from these individuals, or at least some of the individuals. Also, they could have stimulated cells and investigated immune profiles.”
Response: We agree it would be preferable to confirm our data with focused studies on immune cell profiles either by flow cytometry of CyTOF. Unfortunately, we do not have access to PBMCs for these patients or a comparable cohort of AA patients and controls right now and thus are not able to include it in the current study, though it would be a great avenue for future studies. We are encouraged by the fact that we do see a clustering of AA patients and first-degree relatives away from unrelated controls in terms of gene expression from our prior published work (as outlined in lines 403-405).
Concern: “The discussion is also written in a good way, but I find to be "too large" for this study, which should have been more a brief report… Discussion too long for this kind of study.”
Response: We agree that the discussion is fairly long for a short report. However, after careful review, we feel that the points addressed in the discussion are needed to fully support our assertion of an inheritance-specific immune dysregulation in AA and we have trouble identifying truly superfluous sections.
Concern: “Please perform cosmetic changes to Table 1 (i.e., write out words, explain abbreviations, be consistent regarding genetic annotation, etc.)”
Response: We have improved Table 1 by writing out the full words for and/or explaining abbreviations. We have left “AA” as an abbreviation as it has been defined in the text.
Concern: “Do not report the results of the study in the Introduction”
Response: Thank you for this recommendation. After going through the introduction again, we agree that the section where we preview and summarize our findings is not needed and thus have removed it.
Concern: “Do not write results in Fig legends. Here, it should only be explained what the Fig. describes.”
Response: We have removed results from the Figure legends and included them in the text.
Concern: “Line 131: Lack of "alpha"
Response: We apologize for this oversight. We have added the alpha back in the text.
Concern: “The Figures: Would like to see all points in graphs, not just the mean/median”
Response: Thank you for this suggestion. We had previously chosen box and whisker plots to better show the distribution (including median and upper and lower quartiles) as well as clearly identify outliers, one of which is of particular importance for this study. However, we agree that the presentation of all data points is visually easier to interpret for the reader and may ultimately be the better format. Thus, we have replaced all box/whisker plots with dot plots generated by Kaleidagraph.
Concern: “Fig 3: Too small text to read.”
Response: Thank you for pointing this out. We have increased the size of figure 3 overall so that the labels can be larger without causing any confusion as to which row/column of the heat map the text corresponds with.
Reviewer 2 Report
This is an interesting paper underlying the role of IL23/IL17A axis in the development of alopecia areata. This study provides new insights into this hard-to-treat disease. Despite the data are convincing, I am really curious why blockade of IL23/17 axis is not used in the treatment of AA. Furthermore, did the authors looked also fot IL17F?
Author Response
Thank you for your insightful comments.
We had previously considered looking at several of the six IL-17 isoforms, but ultimately settled on IL-17A since it is the most abundant and best described isoform. We acknowledge that IL-17F has a high sequence homology to IL-17A and has been shown to be of relevance in autoimmune conditions such as asthma. However, previous studies in adjuvant models of autoimmunity (including experimental autoimmune encephalitis and arthritis) suggest that IL-17A has more of a pathological role in autoimmunity than IL-17F, and is also a more potent cytokine; though IL-17F is required, along with IL-17A, for mucosal immunity (Dubin and Kolls 2009 Immunity 30, p.9: Interleukin-17A and Interleukin-17F: A Tale of Two Cytokines).
As you point out, IL23/17 blockade is a very interesting avenue to pursue in the treatment of AA, however, thus far there is limited evidence treatment success. Four studies have explored using the IL-17 blocker secukinumab (Guttman-Yassky et al. 2018 Arch Dermatol Res. 310(8):607-14) or the IL-12/-23 blocker ustekinumab (see references below), though with limited success. At the same time, the secukinumab study was limited by low statistical power and low enrollment, though some studies with ustekinumab have had success. Our data clearly supports a role for this pathway in AA and thus ultimately warrants larger studies targeting at blocking the Th17 pathway, potentially in combination with Th1 blockade, to better assess clinical efficacy.
Aleisa A, Lim Y, Gordon S, Her MJ, Zancanaro P, Abudu M, et al. Response to ustekinumab in three pediatric patients with alopecia areata. Pediatr Dermatol. 2019;36(1):e44-e5. Epub 20181018. doi: 10.1111/pde.13699. PubMed PMID: 30338558.
Guttman-Yassky E, Ungar B, Noda S, Suprun M, Shroff A, Dutt R, et al. Extensive alopecia areata is reversed by IL-12/IL-23p40 cytokine antagonism. J Allergy Clin Immunol. 2016;137(1):301-4. Epub 20151120. doi: 10.1016/j.jaci.2015.11.001. PubMed PMID: 26607705.
Ortolan LS, Kim SR, Crotts S, Liu LY, Craiglow BG, Wambier C, et al. IL-12/IL-23 neutralization is ineffective for alopecia areata in mice and humans. J Allergy Clin Immunol. 2019;144(6):1731-4 e1. Epub 20190827. doi: 10.1016/j.jaci.2019.08.014. PubMed PMID: 31470035; PubMed Central PMCID: PMC6900443.
Reviewer 3 Report
In this study the authors have examined AA cytokines by ELISA and have included unaffected family members as well as non-related unaffected controls. Several studies have examined these markers before although there is inconsistency within the literature. In addition, the inclusion of unaffected relatives is an interesting aspect of this report.
The part I found most distracting/confusing was the reclassification of one individual from healthy to AA. In particular, in figure 1A (IL17A) it states that one of the controls has been reclassified from control to AA but then just below in figure 1B (IL23) it says this individual was excluded from that analysis. I am finding this rather confusing so could the authors please clarify whether that individual is excluded or if their data is being moved to another group and if one approach is being applied consistently?
I dont really understand the rationale for moving this individual from control to AA, since the hair loss only developed 12 months later and at sampling they were phenotypically similar to all the other healthy controls. It may be that some cytokine changes occur early (i.e. before hair loss) and others much later, so this particular individual may well fit somewhere in between the AA and healthy groups. I totally accept that this individual probably needs to be removed from the analysis, but it may be less confusing if it is explained in the methods section that one healthy individual subsequently developed AA and so was excluded from cytokine analysis and then just report Figures 1 and 2 following any exclusions. They could still be included in Fig 3 since this comparison is quite neat.
The figures themselves need larger labels as it is very difficult to read them and in particular Figure 3 is very hard to read.
Also in the figures, I was unclear about the red "outlier" symbols as these are not being explained anywhere within the figure legend or the methods and initially I thought this was a significance value. Can the authors clarify if these outliers were removed from the full analysis or not? If so, then what is the specific justification to remove them other than they appear to be an outlier? If they are included within the analysis, then I dont understand why only these individual data points are shown on the plots, since it would seem best to either include all the individual data points or none of them.
Author Response
Thank you very much for your time reviewing this paper and for leaving us comments. We have gone through everything you recommended and addressed each concern individually below. All changes within the text are colored in red for your convenience, which will be removed later.
Concern: The part I found most distracting/confusing was the reclassification of one individual from healthy to AA. In particular, in figure 1A (IL17A) it states that one of the controls has been reclassified from control to AA but then just below in figure 1B (IL23) it says this individual was excluded from that analysis. I am finding this rather confusing so could the authors please clarify whether that individual is excluded or if their data is being moved to another group and if one approach is being applied consistently?
I dont really understand the rationale for moving this individual from control to AA, since the hair loss only developed 12 months later and at sampling they were phenotypically similar to all the other healthy controls. It may be that some cytokine changes occur early (i.e. before hair loss) and others much later, so this particular individual may well fit somewhere in between the AA and healthy groups. I totally accept that this individual probably needs to be removed from the analysis, but it may be less confusing if it is explained in the methods section that one healthy individual subsequently developed AA and so was excluded from cytokine analysis and then just report Figures 1 and 2 following any exclusions. They could still be included in Fig 3 since this comparison is quite neat.
Response: Thank you for the careful review of our data and we apologize for not being clear on this point.
We completely agree that that our data suggest that cytokine changes happen early (before disease is clinically obvious) and that control individual #166 fits somewhere between AA and healthy groups.. We realize now that our original wording in the legend to Fig. 1B (stating that individual CR166 was reclassified) was wrong and we have changed that to say the individual was excluded from a later re-analysis of the data (after we realized this individual had developed AA). In response to another reviewer, we have removed specific results out of the figure legends and describe this fact instead in lines 207-209 of the result section.
To clarify and address your concerns regarding Figure 1A and 1B, control #166 is included in the control group in both figures (and the entirety of the study). However, after we learned this control ultimately developed alopecia, we became curious at how this might affect our results, and out of curiosity reanalyzed some of our data after exclusion of control #166. As you noted, we wrongly stated in the original figure legend that control #166 was reclassified into the AA group, when we should have said that #166 was excluded from the control group for the reanalysis. This error has been clarified in the revised submission. Our intention was merely to show how control #166 is likely showing early cytokine changes and how the trend in IL-23 levels become significant when you remove this control showing these (likely disease predictive) changes. We have altered the figure legend as well as included additional clarification within the methods and results sections (lines 118-20, 277-278). For your information, while checking over this analysis, if we were to reclassify this control into the AA group, the p-value would be 0.033, highlighting that this control’s cytokine levels are more in line with patients already diagnosed with AA. However, we agree with your assessment that this individual likely falls somewhere in between and do not think a full reclassification of this control is warranted. Thus, we only reported the re-analysis of exclusion in our paper, but mention what the reclassification would be in this response to give you a fuller picture,
Concern: The figures themselves need larger labels as it is very difficult to read them and in particular Figure 3 is very hard to read.
Response: Thank you for this suggestion. We have decided to enlarge the figure as a whole so that the figure labeling can be larger without losing the association with particular rows in the heatmap.
Concern: Also, in the figures, I was unclear about the red "outlier" symbols as these are not being explained anywhere within the figure legend or the methods and initially I thought this was a significance value. Can the authors clarify if these outliers were removed from the full analysis or not? If so, then what is the specific justification to remove them other than they appear to be an outlier? If they are included within the analysis, then I dont understand why only these individual data points are shown on the plots, since it would seem best to either include all the individual data points or none of them.
Response: Thank you for noting we needed to be clearer here, and we understand your concern. Our reasoning for showing only these individual data points, rather than all data points was to outline that there were in fact other additional outliers outside of CR 166. None of the statistical “outliers” (determined to be +/- 2 standard deviations from the mean) was ever removed from the main analysis. However, in response to you and reviewer #1, we have decided to present the data as dot plots (showing each value) rather than box and whisker plots, thus removing the necessity to indicate statistical outliers. We still highlight individual CR166 which developed AA at a later time point. Control #166 was the outlier only in the groups for IL23 and IL17A, and the other outliers without a distinction are not from control #166. We hope and feel that the data is now presented in a clearer way.
Round 2
Reviewer 1 Report
Thank you to the authors for considering my comments. I think the article has now been improved significantly.
Reviewer 3 Report
The authors have clarified their methods and my previous concerns are now resolved. I have no further questions about this study.